# Benefits of Implementing Eye-Movement Training in the Rehabilitation of Patients with Age-Related Macular Degeneration: A Review

**DOI:** 10.3390/brainsci12010036

**Published:** 2021-12-28

**Authors:** Anis Hilal, Mazen Bazarah, Zoï Kapoula

**Affiliations:** 1Department of Ophthalmology, Skåne University Hospital, Kioskgatan 1, 22285 Lund, Sweden; 2Laboratoire IRIS, Physiopathologie de la Vision et Motricité Binoculaire, CNRS FR2022 UFR Biomédicale Université de Paris 45, rue des Saints-Pères, 75006 Paris, France

**Keywords:** age-related macular degeneration, eye, central scotoma, macular degeneration, eye tracking quality of life, VFI tools, eye-movement training, neuroplasticity

## Abstract

Age-related macular degeneration (ARMD) is one of the most debilitating eye-related illnesses worldwide. Eye-movement training is evolving to be a non-invasive, rapid, and effective method that is positively impacting vision and QoL (quality of life) in patients suffering from ARMD. This review aims to highlight why a greater adoption of eye-movement training in the clinical and research setting is of importance. A PubMed and ResearchGate search was performed for articles published between 1982 and 2020. Patients with advanced ARMD tend to experience a diminished QoL. Studies regarding eye-movement training for patients with central vision loss revealed overall significant improvements in reading speeds, fixation, and saccade performance. They also experienced less fatigue. In select studies, eye-movement training revealed an improvement in binocular vision, fixation, reading speed, and diminished reading exhaustion. The process of eye-movement training used in some of the studies was rather empirical. The latter requires standardization so that a uniform and applicable methodology can be adopted overall.

## 1. Introduction

Age-related macular degeneration (ARMD) is the most common cause of irreversible blindness in modern industrialized countries. It is a degenerative disease that impacts primarily the macula. Destructive changes involve the retinal pigment epithelium (RPE) and drusen formation (extracellular deposits between Bruch membrane and RPE). The macula is roughly a 1 mm patch of retina that is primarily responsible for sharp and clear vision. This illness has two main forms—the dry or atrophic form, and the wet or exudative form. Both variants impact the macula which is responsible for the sharpness and clarity of our central vision [1,2]. ARMD can render its sufferers with detrimental visual and overall quality of life consequences [3,4]. Most of whom develop ARMD are 50 years of age and older. The risk gradually increases to more than three-fold among those 75 and above. Age, race, and heredity are significant risk factors. In the US, 50% of those that develop ARMD are white, 15% Hispanics, and 2% African American. In the year 2020, it was estimated that shy of 196 million people internationally are affected in varying forms of intensity and manifestations by ARMD [4,5]. Though the advancement of therapies for both has led to significant milestones, those suffering from an advanced stage of the disease can struggle to adapt. Some patients have experienced a dramatic shift in their lives, where work, social interaction, watching TV, or reading a newspaper have become challenges [5]. The limitations implicated by the disease on may vary. Those with milder forms of ARMD notice it but tend to continue unhindered with their daily lives. Others, however, had to endure massive shifts in their lives [6,7]. It can encompass changes such as psychological distress in competition of visually demanding tasks, seeking visual rehabilitation, changing careers, and retrofitting living spaces with visual aid devices. Dependence on social workers for daily tasks may increase, which slowly robs individuals from feeling independent. By proxy, this diminishing in QoL also ensues greater social and financial burdens on the social and health care sectors [8,9]. Patients with central visual scotomas tend to involuntarily utilize their peripheral visual fields to compensate. The latter entails head-turning and side-gazing to align the image with the patch of the retina that is receptive to the image and provides the sharpest vision [9]. This is difficult to achieve due to variables such as constant eye movements, head, and body positions to find the sharpest image, and lack of training to do so. Unlike the fovea where the image naturally converges, head-turning and side-gazing lead to the image falling on random retinal loci instead of one [10]. This creates a dilemma in reading and tasks requiring shifting focus between different objects since the eyes use extra effort and time to do so. Eye movement is naturally controlled by six extraocular muscles that surround each eye and work together in a pulley-like fashion to direct gaze towards the intended direction. This enables the eyes to lock on to individual aspects of an image, accommodate to the distance of the viewed image, follow or track targets, react to sudden stimulus and have salient motions for smooth transition such as in reading. Eye movement aligns the visual axis so that the image will fall on macula [10]. These movements are classified as either saccades (instantaneous rapid movements), smooth pursuits (used to track objects in visual field), vergence (simultaneous rotation of the eyes to maintain the viewed object in focus), and vestibulo-ocular movements (movement triggered by the vestibular system to stabilize gaze during head rotation) [11]. Eye-movement training on the other hand, aims to help patients with macular degeneration locate and utilize a peripheral retinal locus instead of that of the macula. It aims to help patients find the clearest image whilst minimizing search time and effort. Effort is reduced consequentially when targets are located easily and quicker, thus, decreasing head movements and unhealthy postures [10,11]. Eye-movement training is a noninvasive, economical, and progressive approach that is gaining traction among researchers, practitioners, and technology companies worldwide [11]. Currently, there is no consensus regarding the ultimate eye-movement training paradigm is to adopt. Results from the studies in this review aim to shed light on the potential of instating eye-movement training in protocols of visual rehabilitation for patients with central visual field loss.

## 2. Materials and Methods

A PubMed, ResearchGate, Frontiers, and IOVS search was performed for articles published from 1989 to 2020. The keywords mostly used were age-related macular degeneration, eye, quality of life in ARMD patients, eye tracking software, eye-movement training, peripheral retinal locus, and central scotoma. Case reports, original articles, and reviews were considered. Abstracts and non-English articles were not considered for this review.

## 3. Results

### 3.1. Age-Related Macular Degeneration and Quality of Life

Age-related macular degeneration (ARMD) is a disease that in advanced stages involves a detrimental loss of central vision. Patients affected by dry ARMD tend to experience a slow-progressing disturbance that evolves over years or months. Most commonly, they describe their initial symptoms as a slight-to-modest blurriness either when reading words up close (e.g., mobile phone, book) or at a distance (e.g., TV, billboards) [12]. Over time, it may progress to metamorphopsias where sufferers start to see straight lines as crooked and letters looking chopped off, crooked, or blended together [13,14]. The symptoms of crooked vision and rapid decline in visual acuity occur within days rather than years. It involves the ingrowth of new leaky choroidal vessels underneath the RPE leading to exudation and bleeding. Nowadays, 90% of wet ARMD patients experience stabilization and improvement of their visual acuity (VA) by receiving anti-VEGF at the appropriate time [15,16]. With the disease manifestation in mind, add to that the constant intervention of injections and follow-ups. These represent a strain on the quality of life (QoL) of most patients in one form or another [16]. In a 2015 article published by Ord et al. from the University of Utah, the severity of vision loss in 26 ARMD patients (5 males—19%, 21 females—81%) was assessed based on the combination of several objective measures [17]. These include visual acuity, contrast sensitivity, and the functionality of being able to perform everyday vision-related tasks. The study considered how individuals with similar clinical features can demonstrate a broad spectrum of individual-based QoL. Factors that come into play include, but are not limited to emotional, physical, socio-economic, and mental status. A Wozniak et al. article published in 2011 assessed ARMD patients’ QoL using the National Eye Institute Visual Functions Questionnaire (NEI VFQ-25). The study involved 100 patients with ARMD versus a study group comprised of 30 volunteers devoid of any ophthalmic diseases and of similar age and sex [18]. The patients in this study had been treated with anti-VEGF. The results of Wozniak’s article describe a statistically significant difference in QoL of the control group (score = 83.7 +/−11.7) versus the ARMD group (score = 51.1 +/− 20.5) with a *p* = 0.001. The ARMD patients revealed that their disease had rendered their lives with diminished independence and a lower state of acceptance in contrast to the control group. Psychological and social aid were appreciated by the patients to ease their burdens of accomplishing daily tasks. In another study, Matamoros et al. sent out questionnaires to exudative ARMD patients with an inclusion criterion that they had received at least one intravitreal injection of Ranibizumab within the last 6 months [19]. A total of 1888 surveys were sent out and 611 (32.4%) were returned. Among those, 467 met all the inclusion criteria, and out of those, 416 met both the QoL and cost criteria. Those that met the criteria had a mean age of 78.0 years. A total of 159 (34.0%) of them had a VA < 0.5 (20/40 on Snellen chart in feet) on both eyes, 23 (4.9%) had VA < 0.1 on both eyes, 52 (11.1%) had VA > or equal to 0.5 (20/40 on Snellen chart in feet). Both eyes were involved and the patients in the study had endured ARMD for 7 years in the first eye and 2.3 years in the second. It was a cross-sectional and observational survey. Results of the Matamoros paper revealed that mental health, driving, and role difficulties had highlighted the patient’s complaints and restraints on their current QoL. Their QoL had become dependent on their visual acuity, social services that visited them at home and provided them with their needs, and home healthcare services. The patients that had experienced an improvement in their VA had consequently an improvement in their QoL scores. In another study conducted by the Lithuanian University of Health Sciences in 2012, researchers aimed to evaluate the quality of life of 140 subjects using the VFQ and Hospital Anxiety and Depression Scale (HADS). The group with ARMD patients included 70 individuals (56% women and 44% men). The mean age was 68 years. Their results were compared to 70 control subjects (40% women and 60% men) with an average age of 61 years (SD, 5.3). There was an age difference significance in the comparison between the two groups (*p* < 0.05). Their study revealed that ARMD significantly impacted QoL in aspects of general health, vision, dependency, and role difficulties faced by the sufferers in attempts to complete their daily tasks (*p* < 0.0001). In comparison to those with monocular involvement, there was increased susceptibility to symptoms of diminished mental health (r = 0.326, *p* = 0.02) and elevated dependency (r = 0.340, *p* = 0.02) in those with bilateral ARMD (*p* < 0.05) [20].

### 3.2. Eye-Movement Training

The fovea centralis, where 200,000/mm^2^ of cones reside, allows humans to see the sharpest vision within 2 degrees of the visual field. Keeping in mind that both eyes must remain in coordination when it comes to gaze fixation, saccade initiation, interval, and amplitude, the task becomes significantly more complicated [21]. The vestibular-ocular reflex (VOR) and the optokinetic nystagmus (OKN) are low reflexive circuits that help maintain the stability of our vision [21]. The VOR helps the eyes to maintain fixation regardless of head rotation. The OKN is responsible for triggering the quick-follow reflex when an object enters the field of vision [21]. In patients suffering from macula-related disorders such as the case with ARMD, the central vision becomes distraught [22]. The result appears to be compensatory maneuvers by the patient such as gazing from the side of their eyes and rotating their head in a certain manner to see clearly. The peripheral retinal cone cells are more dispersed and rod cells are interspersed between them. Additionally, the cortical area allocated to the peripheral retina is substantially less than that allocated to the macula and, thus, central vision [22]. The preferred retinal locus (PRL) is the residual unaffected retina that the patients with central visual loss tend to rely on. The PRL varies between eyes and patients alike. However, as various studies have demonstrated, due to inadequate adaptation of the eye movements regardless of the timespan a patient had been affected, the PRL might not be the point with the highest visual acuity in the residual, unaffected retina. Without consecutive training to stabilize the PRL into the most fixed point possible, the eyes would resume undulating in their saccadic motions to find sharpness as the patient rotates their head to aid in the process [23]. The result, unfortunately, is not an improvement in reading or vision stability since the eye is not being trained to adapt correctly. As long the patient rotates their head and their eyes are unsuccessfully able to fixate, progress in stability and improvement in reading ability is poor.

### 3.3. Reading Performance and QoL Improvements

In this remarkable 2003 study by Nilsson et al., authors at Linköping University utilized eccentric viewing training to reveal a significant improvement in the reading speeds for 90% of their subjects that suffered large, absolute central scotomas due to advanced ARMD. The study involved 20 subjects (16 female), mean age 77.4 ± 6.0 years (64–86 years) with advanced ARMD (mean VA 0.042 ± 0.016 or 20/475 on Snellen chart in feet). The eye with the worst visual acuity was chosen and none of the patients had undergone any visual rehabilitation before the study. Due to their extremely low VA, none of the patients had been able to read at the time they were enrolled in the study, and none had prior access to magnification devices. A scanning laser ophthalmoscope was used for delineating each subject’s scotoma and detecting their PRL. The PRL was detected by having the patient find and fixate on a letter outside their respective scotoma. Prior to training, 11/20 patients had a PRL to the left of the visual field scotoma. Six out of twenty patients had a PRL located superior to and to the left of the retinal lesion. Two out of twenty patients had a PRL outside the lower left of the retinal lesion, and one of twenty had their PRL to the right of the lesion. Utilizing their current PRLs prior to training with the TRL, they had exceptional difficulty reading (mean reading speed 9.2 ± 6.6 wpm).

The SLO presented the patients with scrolled text in the horizontal direction, right to left and magnified 8–15 times using 32–60 D lenses. Most fixation points were located above and below the retinal lesion. To determine the area with best eccentric viewing otherwise known as the trained retinal locus (TRL), patients had to shift their gaze upwards or downwards until they could read at least four letters at the same time. The SLO then saved the eccentric viewing angle as the TRL. Reading training and text presentation was performed at the same eccentricity. Each session was roughly 1 h, and a mean 5–7 h of formal training was conducted.

Initially, patients tended to direct their gaze back to their original peripheral retinal loci before they are guided back to their respective TRL(s) by the low vision therapist. Magnification lenses are then introduced, and patients were asked to read the text displayed aloud for a period of 3 min. Initially, the screens displayed help lines (long horizontal lines) that the patient would focus on. These lines were located at the exact eccentric viewing angle respective to each patient’s TRL. Training books with custom help lines for each patient were provided for homework during the week. Eventually, the SLO training sessions were without help lines and the system would detect if fixation fell short of the TRL. During the sessions, the system could pick up on the patients reading faster and using a microphone detect whether the words were said correctly. Different words would be introduced at the same magnification and eccentric viewing angle unique to each subject. Twelve out of eighteen patients who were able to learn eccentric viewing had their TRL above the retinal lesion (below the scotoma). The remaining 6/18 patients had their TRL below the retinal lesion (above the scotoma). The average angle of eccentricity measured by the SLO was 7.8 ± 2.0°.

The results revealed eccentric viewing to be possible in 18/20 of the patients (90%). The two other patients experienced difficulties (one kept using her PRL and the other quit). The 18 subjects who learned to utilize eccentric viewing benefited from significant improvement of reading speeds up to 68.3 ± 19.4 wpm (*p <* 0.001) from mean reading speeds of 9.2 ± 6.6 wpm prior to training. Authors report that individuals of an average age of 77.3 (similarly to the current study) with normal VA had an average reading speed of 82 wpm. [24].

The randomized and controlled 2019 study by Kaltenegger et al. shed light on the effects of home reading training on QoL of AMD patients. It utilized reading training (RT) software that was provided to the participants on laptop computers, Rapid Serial Visual Representation (RSVP) as a visual training method, SLO (scanning laser ophthalmoscopy, model 101, Rodenstock, Munich) for reading speed and fixation assessment, the Montgomery–Åsberg Rating Scale (MARDS) for severity of depression assessment, dementia detection test (DemTect) for cognition, and the Impact of Vision Impairment (IVI) questionnaire for QoL assessment.

Thirty-seven patients (57% women, median age 72 years) with advanced AMD having no statistically significant differences in their baseline VA, age, reading speed, disease duration, and magnification required met the requirements for this study. The subjects were divided in two groups. The first group (*n* = 25) received RT from the start, and a control group (*n* = 12) that received placebo training for 6 weeks using crossword training (30 min, 5 days/week) before switching to RT and joining the first group.

The RT and control groups eventually both received 6 weeks of RSVP training that were divided to 30 min training sessions for 5 days per week. Assessment took place at three stages for the RT group and four stages for the control. The first stage (t0) for both groups was prior to any training taking place to establish a baseline assessment. The second stage (t1) was exclusively for the control group after they had been undergoing placebo training for 6 weeks. The third (t2) and fourth (t3) stages involved an assessment for both groups together right after RT was completed and 6 weeks post-training completion respectively.

Reading training involved text being displayed on the laptop screen as a sequence of single words, one at a time and by RSVP. Text magnification was preset for each individual and based on their clinical assessment prior to beginning training. The training exercises for both groups took place at home. Eye-movement evaluation was performed on 17 of the subjects during the assessment stages (t1–t2). The patient had to fixate on a cross at the center of the screen and read words aloud as the SLO correlated reading speed with fixation on the cross and the PRL while reading single words. The variables measured included: number of forwards and backward saccades, time between text presentation and vocal articulation, frequency of looking at the text and finally the number of fixations.

The results revealed the following:

For the control group, average reading speed increased from 78.7 wpm at t0 to 87.4 wpm at t1, thus, an average 8.7 wpm increase was noted. Afterwards, as the control group joined the RT group and began RSVP training, an average 10.6 wpm increase from t1 to t2 was detected. Eventually, a 6.0 wpm increase from t2 to t3 was noted after 6 weeks of training conclusion.

According to the authors, a statistically significant change from t0 to t2 occurred.

For the RT group, a statistically significant increase in reading speed took place between each of t1 (69.4 wpm), t2 (82.6 wpm), and t3 (85.0), respectively. Fourteen (38%) of the thirty-seven patients developed an average ≥ 10 wpm increase in their reading speeds by the end of the study, and three patients (8%) encountered a ≥10 wpm decrease. The authors noted interestingly that the patients with initially lower reading speeds gained more improvement towards the end of the experiments. Between t2 and t1, the other tested parameters—VA, age, disease duration and magnification—were not directly correlated to reading speed improvement. At t1, however, reading speed revealed a direct correlation with VA and disease duration. A negative correlation between reading speed with magnification requirement was also noted at t1. At all time intervals, fixation stability while fixating at a cross was not statistically different in both groups, revealed no significant changes as time progressed, and showed no correlation with the increase in reading speed. Most patients fixated with a retinal area above the retinal lesion with a 6° radius. There was no significant change in PRL throughout the training period.

Emotional status assessment using MARDS for depressive symptoms revealed a statistically significant difference in improvement for the RT group at t1–t2 versus the control group at t0–t1. The QoL IVI assessment revealed a significant improvement for the RT group at t1–t2 while the control group evaluation remained unchanged between t0 and t1. The cognitive status test DemTect showed no changes during the training. The authors concluded that RSVP training for AMD patients who already use magnification aids, yields statistically significant improvements in reading speed and QoL while contributing to prevention of depressive symptoms [25].

Though the Coco-Martin et al. study was conducted on patients with various macular degenerative disorders and not ARMD, it studied the impact of a reading rehabilitation program (RRP) on their QoL, and it was important for us to be included in this review. The study included a total of 36 patients, 17 of whom suffered from Stargardt’s disease (STGD), 11 with adult-onset foveomacular vitelliform dystrophy (AFVD), and 8 patients with myopic macular degeneration (MMD). Mean VA was as follows: 0.57 ± 0.38 (STGD) or about (20/30 on Snellen chart in feet), 0.51 ± 0.38 (AFVD) or about (20/40 on Snellen chart), 0.49 ± 0.24 (MMD) or about (20/40 on Snellen chart), without low-vision aid. Mean VA with low-vision aid was 0.89 ± 0.20 (STGD) or about (20/25 on Snellen chart), 1.08 ± 0.17 (AFVD) or about (20/20 on Snellen chart), 0.99 ± 0324 (MMD) (20/20 on Snellen chart). Controls had the same diseases (five patients with STGD, five with AFVD, five with MMD) as the those undergoing RRP. Mean VA for the control group was 0.55 ± 0.25 or about (20/40 on Snellen chart) without low-vision aid. The goal of the training was to evaluate the changes in reading speed, duration, and font size in each in-office session. The reading rehabilitation program included 4 in-office sessions and 39 in-home sessions where patients would be training over a period of 6 weeks. A short version of the World Health Organization QoL questionnaire was used prior and post RRP training. Results revealed a significant improvement in reading speeds for the patients in the RRP group (*p* ≤ 0.01) over that of the control groups in the tested reading parameters. Patients with STGD in the RRP group reported a greater improvement than others in the same group when it came to QoL, though, all reported an improvement, nonetheless. In comparison, QoL assessment of the control group did not reveal any significant improvement in any of the parameters questioned. The authors concluded that RRP could significantly improve the reading performance of their patients with central vision loss and consequentially improve their QoL [26].

In the following Seiple et al. study, eye-movement training was conducted on 16 patients that suffer from ARMD. Their mean age was 77 at the time. During the training, the patients could not rotate their heads and one eye was trained at a time. An eye-tracking system was used (Model 504 Pan/Tilt; Applied Science Laboratories [ASL], Bedford, MA, USA) in addition to a head tracking system. The study used the bright pupil-illumination technique for its eye-tracking system. It encompassed tracking the pupil and the ensuing corneal reflection known as the Purkinje image. Due to the difficulties that patients with ARMD face in localization and fixation of an image, the researchers had to calibrate the eye tracker specifically. This was done by allowing the patient to identify and fixate on letters with a reference acuity size that they set for themselves. This allowed the probability of identifying the stimuli with their PRL higher. The 8-week training program incorporated successive increases in the difficulty of the task performed during the sessions. Initially, the subjects were tasked to practice horizontal saccades before graduating to letters and full words before ending the 2-month program with training in reading complete sentences. The training was essentially focused on eye movement and fixation.

The results were promising and revealed an increase in reading speeds by an average of 25 words per minute (wpm) over the reading speeds they had originally (*p* < 0.001). This was of particular interest to the researchers at the time since the subjects had received limited training in reading whole sentences. The 25-wpm speed bump allowed the patients to reach the near-normal reading speeds of an average individual with no ARMD. Based on the results they attained, it is estimated that after their training program an individual with ARMD could read a 2000-word newspaper article 5 min faster. However, they could not detect a significant improvement in visual acuity after the program was completed. [27].

### 3.4. Oculomotor Response and Fixation

Though the Van der Stigchel et al. study was conducted on patients with Stargardts disease and not ARMD, the findings and concepts utilized for better understanding central scotomas in macular degeneration made it important for us to be included in this review. In their 2013 study, Stefan Van der Stigchel and Richard Bethlehem compared the oculomotor response of patients with macular degeneration (MD), particularly Stargardt disease to that of control individuals and those with a simulated central scotoma [24]. The experiments compared search latency, saccade amplitude, number of saccades to target, intersaccadic interval, and saccade direction. The study included four patients that suffer from central vision loss, ten healthy controls (age 29.9 ± 10.0/four males) for the visual experiment and five controls for the same task but with simulated scotoma (25.2 ± 1.3/3 males). Eye movements of were registered for the subjects’ dominant eyes using the Eyelink 1000 infrared tracker that is constantly calibrated prior to experiment conduction. Patients watch a computer screen at 57 cm from their eyes and focus on certain calibration points during the experiment. The authors assumed that patients were utilizing their PRLs during the calibration since they were fixating on a particular point with same retinal location. During the trials, distractions and intended visual targets were displayed on the screen. A visual field test was conducted to better comprehend the degree of the central scotoma suffered by patients with macular degeneration. A 1.5-degree circle was projected on the screen and could appear in 33 locations. As the targets would appear, participants would maintain their gaze on a fixation cross and press keys to illustrate if they had seen the target. A total of 146 trials were conducted. The visual field testing revealed partial scotoma in all patients with macular degeneration. Additionally, Scanning Laser Ophthalmoscopy (SLO) was utilized to measure fixation stability and absolute locus of the PRL in patients suffering from macular degeneration. SLO revealed that patients were using peripheral fixation. For the visual search task, subjects had to locate and determine the orientation of the letter C in a maze of distractors as fast as possible for a total of 110 trials. Healthy controls in the simulated scotoma group had to participate in the task as well. The results revealed that in comparison to both the control and simulated group, MD patients had a longer search latency (*p* < 0.001); otherwise, the time it took the MD patient to find the target and have it in focus. Interestingly, when it came to saccadic amplitudes, it seems that those suffering from macular degeneration had smaller saccadic amplitudes when their gaze was directed towards the scotoma in their visual fields. Additionally, MD patients required more saccades to find a fixed point and their intersaccadic intervals (time between when one saccade ends and the next is initiated) were longer than both the control and simulation groups (*p* < 0.01). Furthermore, the notion that MD is attributed to solely a central scotoma does not appear to be the case in the latter study. Rather, the borders of the scotoma would protrude into the peripheral visual field thus rendering the term “relative scotoma”. The researchers could prove, using an eye tracker, that the PRL for MD patients was not a stable point. This illustrated that the eye took a longer time to align the image on a point on the retina where it be rendered the sharpest. In the case of MD, the macula is damaged and can in severe cases no longer serve as the locus of sharp vision. Thus, more saccades to find a fixed point could be an adaptive phenomenon to locate a patch of well-functioning retina (PRL) that renders a better VA [28].

The Janssen et al. study entailed the use of eye-movement training for nine participants (four males, five females, aged 52–90) with central field loss (CFL) caused by ARMD. The study aimed to train each of these individuals to detect the location of their scotoma and use their PRL to uncover concealed information. The hypothesis of Janssen et al.’s study was to train subjects to use their PRL to saccade towards the area concealed by the scotoma. Three research questions were proposed and evaluated. The first was an improvement in the efficiency of locating the scotoma. The second was in correctly identifying the concealed figures and the time required to do so. The third regards whether the eye-movement training protocol led to an improvement in the performance of other daily life tasks. PRL eccentricity and its location for each person were established by microperimetry, scanning laser ophthalmoscope (SLO), and binocular scotoma mapping.

The experiments involved asking each subject throughout the trials to focus on a central fixation point on the screen whilst their scotoma covers the target image. Some experiments entailed two images in the visual field. The first image was concealed by the scotoma and the second image presented diametrically opposite in the visual field and visible. The tasks involved asking the subject to locate the image veiled by the scotoma by using the visible image as a reference. The other task involved locating the concealed image without a reference silhouette. A clock test was also conducted where patients were asked to focus on the center of the clock and report on any missing numbers in their visual fields. Further tests included measuring binocular visual acuity using the MNREAD chart.

A total of 480 trials were done over a cumulative period of 6 h. Ten blocks were allocated to everyone, whereby each block entailed 48 trials and was spread over 2 weeks. The results revealed no significant difference in saccades’ performance towards locating the scotoma as the time required by the experiments to locate the scotomas shortened and the images concealed under the scotomas diminished in size. Six of the nine participants revealed faster saccades after training. Additionally, four participants could maintain this performance when a retention test was conducted after 2–3 months after training was concluded. The authors discovered that the subjects who benefited the most from training had their scotomas in their upper visual fields. Two-thirds of the participants revealed a degree of benefit from training in aspects of conducting faster saccades and in saccades awareness. However, the consensus among the authors was that there was no significant transfer of benefits (from training) to another task. Additionally, there was no significant improvement in accuracy in reporting the number of blobs in each scene. The study admits to its limitations that include only 6 h of training, the inability for the subjects to move their heads, and pushing the subjects to move their eyes quickly in a short time frame [29].

The following Léne et al. study aimed to explore the changes in eye movements in early ARMD by inducing an artificial scotoma is induced in otherwise healthy patients. Eye movement and eccentric visual function adaptive behaviors were mainly studied.

The study involved fifteen subjects (7 males) aged between 19 and 25 (M = 21.69, SD = 2.13), all of whom had either normal or corrected visual acuity. Eye movement tracking took place using the Eyelink 1000 Plus Tower Mount, SR. The experiments were carried out in a series of four blocks that involved 75 trials for each individual daily for a total of 10 days. The baseline trial without any scotoma was established on the first day. In each of the other sessions, the subjects would have the Eyelink system fitted and calibrated so that their gaze was tracked. The Eyelink would then feed the data regarding gaze to the monitor setup in front of the subjects. Based on the foveal information provided by the eye-tracking system, the scotoma would appear on the screen either as a dark circular figure that is 4° in diameter, or an invisible scotoma with similar dimensions. The “invisible scotoma” would mirror the color of the background but block the target that the patient was trying to direct their gaze towards. As the participants tried to find the target using their peripheral vision, they were instructed to press a button on the table to indicate the orientation (clockwise or counterclockwise). The scotoma would disappear when the subject pressed the button. The following saccadic parameters were calculated and assessed: horizonal and vertical final eye position versus the moment the subjects pressed the button to indicate they saw the target and determined orientation. Additionally, horizontal and vertical saccadic endpoint compared the saccade endpoint with where the saccade was first generated. Gaze variability of the final position after each trial was also assessed in addition to the duration and velocity of the first saccade peak and saccade reaction times. The results revealed that he subjects’ responses were similar in the presence of both a visible and an “invisible” central scotoma.

Saccade reaction times increased significantly when the scotoma was initially introduced (*p* = 0.001). As the practice trials progressed, the reaction times gradually decreased and became like those recorded at the baseline trial with a non-significant difference between the two (*p* = 0.116).

Saccade peak velocity results revealed no significant differences in comparison between the final and baseline trials. Button response times significantly improved and shortened towards the final trials in comparison with the prolonged times recorded when first exposed to the scotoma (*p* < 0.001). The subjects’ PRL was overall located in the upper visual field after exposure to the central scotoma.

The authors concluded that the changes in eye movements revealed an improvement in eccentric discrimination after practice sessions with a central induced scotoma [30].

The Morales et al. study’s objective was to see whether fixation stability (FS) of the preferred retinal locus (PRL) could be improved in patients with foveal vision loss when paired with biofeedback fixation training (BFT) with microperimetry [26]. BFT is a form of eye-movement training that incorporates a task-oriented system for behavioral therapy. Patients train by performing eye movements in a specific direction and focusing on a visual target. By doing so repetitively, a selected retinal locus attempts to align itself with the visual stimulus. Concurrently, a biofeedback audio signal in the form of a beeping sound increases in frequency as the visual stimulus and retinal locus approach alignment. The study included 67 patients: retinal geographic atrophy (GA) *n* = 30, moderate dry ARMD *n* = 19, patients with Best’s disease *n* = 9, myopic macular degeneration *n* = 6, and central serous macular degeneration (CSR) *n* = 3. Those recruited had diminished FS and a VA of less than 0.3 LogMAR or about (20/600 on Snellen chart). The objective was to examine whether BFT could improve the FS in these patients. BFT was conducted in conjunction with the MAIA microperimetry that can detect the amount of retinal displacement (P1) within 1° of a set reference point. Additionally, it can measure the bivariate contour ellipse area (BCEA). The latter correlates to 95% of a patient’s retinal loci during an ongoing fixation attempt. Using the P1 and BCEA, MAIA could provide an FS score value. The eye with the better VA was chosen, and if both eyes had similar VAs, that which had the better FS was selected.

Group A patients (*n* = 28, 20 females, mean age 64.7 + 22 years) underwent BFT using their current PRL (assessed by MAIA Standard-Macula Test) as a reference baseline. Meanwhile, in group B (*n* = 39, 27 females, mean age 70.4 + 14 years), instead of relying on the patients’ current PRL, a locus harboring the most optimal functional characteristics was located using MAIA. It utilized a Low Vision-Assessment grid test (30°, 83 stimuli) that grades retinal sensitivity based on its affinity to detect stimuli at four different decibel values. It identifies a retinal locus with “good” or “relatively good” light sensitivity in the horizontal axis. Additionally, the Fixation–Training–Target grid test was used in conjunction to determine a new target locus for the BFT. The new locus was set in the center of two adjacent stimuli bearing the highest light sensitivity and simultaneously the closest from the fovea to the baseline PRL. Ten-minute sessions were conducted 12 times per week followed by a 3-month no-training period after which the training would resume for 1 week. The training entailed asking subjects to direct their gaze slowly towards a target and concurrently audio signals would increase in frequency the closer the desired retinal locus was from the target. Afterward, the subjects were asked to attempt mimicking the steady gaze movements during training in their daily lives when trying to fixate on a visual target. Assessment of results took place 2 weeks post-BFT completion. The mean central scotoma sizes for both groups were 5.7° + 4.5°. The results revealed the following:

FS Index P1(%): It did not improve in 50% of subjects in group A and 18% in group B.

BCEA@95%: 35% of group A showed no improvement. In contrast, 10% of the subjects in group B did not improve.

Mean VA (LogMAR): Improvement was recorded in 16 subjects (57%) of group A and 26 (67%) of group B. No change in VA was recorded in 4 (14%) of group A and 10 (25%) of group B. VA decreased in 8 (29%) subjects in group A and 3 (8%) in group B.

Mean Reading Speed (wpm): See Table 1.

The study then deducted using the Mann–Whitney test that there was no significant difference between the variables of group A in comparing baseline vs. therapy end values. The difference in group B variables (except for light threshold sensitivity) was significant. Except for VA, all other parameters in the outcome revealed a significant difference between groups A and B. The study then concluded that BFT does improve fixation for those with eccentric vision. The authors elaborate on the notion that BFT is based on neuroplasticity. The recurrent stimulation of healthy retinal neural sensors at the spot of highest sensitivity and the shortest distance from the anatomical fovea revealed better results than stimulating the PRL spontaneously developed by the patients themselves. An important point is highlighted regarding retinal cone density and fixation. They refer to previous studies that revealed an indirect correlation between photoreceptor density and the farther one strays away from the fovea, whereby the closer the fixation points are to a retinal meridian having high cone density, the better the visual capabilities become. This might explain why training to fixate at such a point has revealed the significant improvement deducted by this study [31].

A study by Walsh et al. involved simulating a central scotoma in the visual fields of young, otherwise healthy adults. The goal was to reveal the changes in eye movement, fixation, reaction time, and what peripheral retinal locations were chosen by the subjects after 11 blocks of eye-movement training.

A total of 1782 trials took place over 3–6 weeks. Twelve individuals (4 males and 8 females, aged 28.4 +/−u 5.0 years) took part in this study. They had no pre-existing health conditions that could have otherwise impacted their vision and affected the study’s outcome. An EyeLink II eye tracker was used to simulate the scotomas in the visual field and to track eye movement throughout the entire study. The subjects would have to look at a monitor and find the letter O among a multiple letter Cs displayed on the screen with each letter oriented differently. The subjects were separated into two groups. One group was subjected to a central scotoma with soft edges while the other was subjected to a scotoma with sharp edges. Both were reproduced using a gaze-contingent mask (a 10° circular disk in diameter) that creates the scotoma based on the information provided by the eye tracker. The circular disk used was opaque inside and transparent on the outside. The sessions included trials where the O letter was completely absent from the screen. Each of the groups went through the 11 blocks once with the sharp-edged scotoma and once with the smooth-edged scotoma. The results revealed a significant decrease in reaction time from an average of 8.194 s to find the target in the very first block to 3.587 s in the final block. A significant adaptation effect was detected (*p* < 0.0005). In addition, the study revealed a direct correlation between search reaction time and the number of fixations that took place, whereby fewer fixations took place towards the final blocks of the experiment (*p* = 0.002).

The authors concluded that a significant adaptation to a centrally simulated scotoma was possible in their study. They also noted that in the initial sessions, the subjects did not exhibit a select point on their retinas where most of the fixation took place. As the experiments came to an end, a consolidation of fixation points was noted near the border of the invisible scotoma. Interesting findings emerged from the data regarding the last fixation distributions after conclusion of the training blocks. Thirty-six initial fixation distributions were compared with 36 final distributions collected from the 12 subjects in the first and final three blocks of training. The authors observed two tendencies. The first, more uniform fixation distributions (33 out of 36) were recorded in the final three blocks of training in comparison to 21/36 recorded in the first three blocks. This meant that subjects were fixating more uniformly at certain locations after scotoma search training than in the initial stages. The second tendency observed was that final fixations seemed more prone to congregate when subjects were training with a sharp-edged scotoma than a soft-edged one. The fixation distribution after scotoma training seemed to cluster in an arcuate fashion close to the edge of the scotoma in one or two adjacent peripheral quadrants. Interestingly, central distributions decreased from an average of 17 to 6 after training, while peripheral upper-field clustering increased from 7 to 26. Initially, retinal selection for subjects newly exposed to the scotoma was rarely peripheral. Rather, initial retinal selection seemed to favor the location of the scotoma.

As training progressed and authors noticed a trend of last fixation consolidation to a zone adjacent to the edge of the simulated scotoma. This was a particular finding since the stimuli during training avoided favoring a certain location in the visual field or easily giving away the shape of the scotoma. After 11 blocks of training, most participants favored a retinal location nearing the scotoma border in the upper left visual field. The authors stipulated that a possible explanation for choice of the upper left visual field could be that the participants might have a habit derived from reading. Furthermore, the authors noticed that visually healthy subjects challenged with a simulated scotoma were ultimately able to establish a PRL preference faster than in patients with actual central vision loss (CVL) in studies with similar parameters. A possible interpretation could be that CVL patients endure multiple tasks simultaneously in their daily natural surrounding and, thus, finding a PRL with more uniform fixation distribution could take more time. Therefore, the authors suggest early and intensive single-task training in the initial stages of CVL since it might be of therapeutic value in PRL development [32].

In the following Liu et al. study, eight visually healthy participants (aged between 25 and 28 years, four males, four females) underwent a novel eye-movement training protocol in presence of an artificial scotoma. The aim was to investigate whether a random PRL could be trained to improve VA, letter/face/object recognition, spatial attention, and RSVP (Rapid Serial Visual Presentation) reading speed. The study involved creating a false circular scotoma (radius = 2.5°) on a monitor that corresponded to the location of the subjects’ foveal vision. This was done using the video based Eyelink 1000 for gaze tracking and scotoma projection. Furthermore, to push the subjects to use only the PRL that the software randomly had assigned to their visual field, a Gaussian filter blur was applied. The blur obscured the clarity of the background image except for a small circular area (radius = 2.5°) which remained clear was created at random. That area of clarity became the PRL. Thus, every individual was now faced with a central scotoma, a blurred peripheral field, and a randomly created area of clarity. Each subject was required to complete three tasks. Task 1 involved asking the subjects to identify the image projected to the newly created PRL. Three sets of images were projected respectively (faces, objects, and words). Subjects were required to correctly identify whether the faces were male or female, if the objects were cartoons or not, and finally, if the words projected had actual meaning or were simply random letters. Task 2 was calibration. Task 3 involved visual search. In the latter, blurry distractors had been placed in the visual field to make it even more difficult. Subjects were then told to find the clearest image of all, using the PRL that they had previously trained with in Task 1. The study was divided into separate blocks and each block had 30 trials. It took an average of 6–10 h of explicit training to complete all training blocks. The results revealed the Improvement in recognition accuracy by 87% at the end of training (*p* < 0.001). Additionally, increase in search accuracy by 22% (*p* = 0.001) and significant shortening of search time by the end of the training (*p* = 0.01) were detected. Other significant improvements include letter recognition (26% improvement, *p* = 0.02) and RSVP reading speed (18% improvement, *p* = 0.04). The results reflect the values obtained both during and after the sessions were completed. Subjects were asked after completion of blur training to redo the same sessions but with no blur present. The images projected into their non-blurred visual fields were at the same locations (PRL location) as when the blur was present. The results thus reflect a congruity with the final training sessions. The authors could then conclude that with proper training to a randomly assigned PRL, a significant improvement in letter/face/object recognition, spatial attention, and RSVP (Rapid Serial Visual Presentation) reading speed could be achieved. This technique was proposed by the authors to be a possible rehabilitation protocol for those suffering from diseases leading to central vision loss [33].

## 4. Discussion

Macular degeneration can lead to significant impacts on quality of life. Articles published between 2005 and 2019 regarding the QoL of MD sufferers have largely shown a debilitating disease that renders most of its victims having to get accustomed to a different way of life. Eye-movement studies and training programs with the aid of state-of-the-art eye-tracking devices and software are starting to introduce a new paradigm shift. Victims of macular degeneration tend to be 50 years and older. As vision shifts rapidly, especially in the exudative form of ARMD, patients must succumb and adapt to a disturbed visual acuity. Reading script, watching TV and most of all driving are some of the greatest impacted sectors of life by ARMD. Patients seemed to have a difficult time adapting to these conditions over years even after treatment with one or several anti-VEGF intravitreal injections. The overall QoL of the above-mentioned patients appeared to be highly reliant on their VA at the time. Whether it is a case of advanced exudative ARMD or a longstanding atrophic ARMD, neither QoL nor VA improved over the years as the patients grew with age. A destructive change to their VA in terms of ability to drive, read fine print and signs can be detrimental to their socioeconomic state. Eye-movement training employed different protocols to accomplish an overall subtle improvement of visual acuity. The most significant improvements were in letter/face/object recognition, spatial attention, fixation, and reading speeds. In experiments such as those of Nilsson et al., Kaltenegger et al., and Morales et al., there were significant improvements in reading speeds that could be substantial on the QoL of patients.

In Nilsson’s study, reading speed improved from 0-few wpm up to 68 wpm in 90% of their subjects after a total of 5–7 h of training using eccentric angle viewing and a SLO. Eighteen out of twenty patients could reach post-training reading speeds up to 80% of what visually healthy individuals in the same age bracket could achieve. Subjects in the Morales et al. study gained a similar significant reading speed improvement from 56 ± 30 wpm up to 63 ± 36 wpm after concluding their BFT eye-movement training sessions. Ten-minute sessions were conducted 12 times per week followed by a 3-month no-training period, after which the training would resume for 1 week. As authors in both studies interpret, reading speed has much more of a substantial impact on the daily life of an individual in the 70s age bracket. In the above example, two separate studies utilized two separate methods and different patient groups with central visual scotoma caused by damage to the macula. Although patients in the two studies had different starting reading speeds and different training exercises, after training, those patients that learned to use their eccentric vision in a more efficient and tailored manner benefited in a similar fashion. The post-training reading speeds in both studies were relatively similar even though the pre-training speeds were drastically different. Additionally, the time allocated to training differed between the studies and yet similar results were achieved. The Nilsson study even noted a negative correlation with the length and number of training sessions after the 5–7-h threshold is exceeded. Thus, patients benefited significantly in reading speeds when optimal peripheral retinal loci are trained correctly. It is still unclear if the length of training plays a crucial role or not. This is because each study is utilizing different parameters and sometimes equipment rather than attempting to reproduce the same results under similar parameters but with a shorter/longer or frequent/less frequent training sessions. In some studies, the scotomas were simulated for otherwise healthy individuals visually. Using advanced eye trackers, one could create a gaze-tracking scotoma that simulates the early effects of MD. The common goal of all the reviewed studies was to study the impact MD had on an individual’s visual field and whether eye-movement training paradigms could illicit some form of visual improvement in one or more parameters.

Regardless of the training protocol, most studies utilized experiments that included a set of blocks, otherwise known as training sessions. These would include certain visual tasks that would challenge the subject performing the task in various means. Challenges might encompass locating a certain image in the visual field or discerning one image from the other. Others had patients tracking a pattern of lights and attempting to saccade slowly between them. Such challenges were usually performed in blocks that would be repeated over several weeks. Subjects were either patients with MD, controls with and/or without a simulated central scotoma. Subjects would be challenged prior to eye-movement training to set a baseline that post-training results could be compared to.

Irrespective of what paradigm was utilized in training, most results seemed to indicate a certain enhancement in the quality of vision. Examples include improvement in saccadic initiation, amplitudes, fixation, pro-saccadic intervals, reading speed, discerning objects, and generally decreasing the effort required to complete the tasks. This was evident in otherwise healthy subjects with induced central scotomas and patients suffering from central visual field defects due to diseases such as ARMD. Patients with ARMD experienced improved reading speeds and less exhaustion as training elapsed. Results reported better fixation, faster reading speeds, and a decreased tendency for head rotation to compensate for a distraught central visual field. When correlating the length of the training sessions adopted by each study (see Table 2*)* it varied greatly from a few days to several months. Interestingly, regardless of the length of time during which the experiments and training were conducted, various subjects were still revealing improvement whether in reading speeds, saccades, or fixation. In fact, six of the seven studies discussed in this paper revealed a significant improvement in some or all the visual parameters that were being tested. The study by Janssen et al. did not detect significant improvement in the tested visual parameters such as search latency time and lasting improvement in reading speeds after the training sessions were concluded. It questioned whether the 6 h training time each subject received was sufficient. Additionally, their subjects had their heads fixed and could only move their eyes during the experiment which does not represent the reality of how vision is utilized in daily life according to the authors. Authors of the Morales et al. study recommended the adoption and implementation of BFT as a form of eye-movement training method based on the promising results their study revealed. Search latency, fixation, object identification, and reading speeds were among the most common tested parameters in the documented studies. Neuroplasticity was suggested was a possible explanation of the notable improvement in performance of the visual task.

Neuroplasticity is thought to be the mechanism by which the brain forms new connections and pathways to adapt to certain changes in the external and internal environment. As C.A. Nelson claims, evidence is slowly emerging to reveal that visual plasticity can exist beyond the realm of development in childhood. Rather, plasticity can be present whenever changes in the external and internal environments are adequate to stimulate it [34]. A concept termed as the Hebbian competition illustrates how those neuronal cells used most often and those most receptive to stimuli tend to strengthen their connections and flourish, whereas those unused tend to be pruned. This concept is conceptualized when considering cases of amblyopia and diminished VA in childhood due to strabismus, anisometropia, and cataracts. In a Li et al. study in 2011, playing videogames revealed an enhancement spatial vision and an induction of cortical plasticity in the adult amblyopic visual system. Twenty adults with amblyopia aged 16–61 and having VA that varied between 20/25 and 20/480 were included in the study. Their results revealed substantial improvement in a variety of parameters for those who participated in playing action and non-action video games for 40–80 h (2 h/day), challenging the amblyopic eye. Improvements to visual functions (visual acuity, positional acuity, spatial attention, and stereopsis) were respectively 33%, 16%, 37%, and 54%. The study employed a cross cover experimental design that included 20 h of initial occlusion therapy and afterwards 40 h of video game playing time. The authors observed that VA recovery was five times faster than that expected of occlusion therapy in current childhood amblyopia treatment regimes. To evaluate for neuroplasticity, positional noise and modeling was employed. Spatial distortion decreased by 7% and neural processing efficiency increased by 33%. The authors concluded that although this study was of a small sample size, video-game therapy could be one of the therapeutic approaches to childhood amblyopia and other cortical dysfunctions [35]. Perceptual learning is subjecting the brain to challenging and repetitive processes that could allow it to persistently enhance its visual system. It works through tuning of individual neurons in the latter system to improve the differentiation of small differences in two targets. The longer and more repetitive the visually stimulating experience is, the more like it is to be transferred to the long-term memory realm [36]. The lesion projection zone (LPZ) is known to be the area within the visual cortex that has lost input from the damaged retinal area. Diseases that can generate LPZs include ARMD, Stargardts, and Retinitis Pigementosa (RP). In a 2008 study by Masuda et al., functional magnetic resonance imaging was utilized to assess the LPZ and the abnormal cortical signals in patients with juvenile macular degeneration (JMD). The subjects included four patients (two with Stargardts and two with cone-rod dystrophies) and three healthy controls. All underwent experiments that incorporated performing stimulus-based fixation tasks (one-back tasks) consisting of detecting and reporting color change of a fixation dot (using their PRL) and reporting consecutive repetitions of the same stimulus or by viewing a stimulus. In other experiments, the subjects would passively view the stimuli and were not required to perform any actions. Results revealed that large areas in the V1 region of the visual cortex were non-responsive. The latter areas corresponded to the foveal projection zone in the visual cortex and were labeled as the Lesion Project Zone. The study revealed that during passive viewing, the LPZ was silent and did not show any significant activity. During stimulus-related tasks, however, the LPZ showed significant activity that was otherwise not present and could not be induced by passively viewing stimuli. Controls had no significant cortical response in the simulated LPZ induced by having them maintain retinal fixation with their PRL. However, the authors did not believe that variation in eye-movement patterns could be an explanation of their findings. They reason is that the LPZ is a fixed area in the cortex that does not change regardless of how the eyes move. Additionally, eye movements were recorded in two of the four patients and found no significant difference in eye movement with or without performing the stimulus-based tasks. They concluded that visual stimuli combined with particular tasks could drive stimulus-synchronized responses in the LPZ of 3/4 of the JMD subjects [37].

However, a study by Nelles et al. in 2010 studied the impact of extensive eye-movement training on eight patients that had suffered ischemic occipital cortical lesions of the striate cortex. Their visual fields were rendered with a homonymous hemianopsia visual field defect. This study utilized fMRI to detect changes of neuronal activity in the visual cortex and additionally the unaffected extra-striate cortex. fMRI was performed during hemifield stimulation experiments before and 8 weeks after completion of eye-movement training (4 weeks of total training). Prior to training (at baseline), affected visual hemifields were subjected to stimulation and fMRI revealed extra-striate cortical areas. This activation was sustained after training completion. Interestingly, 4 weeks after the completion of training, additional activation of extra-striate cortical areas was detected in comparison to the areas detected at baseline prior to any training. Although no changes in the size of the visual fields were detected, eye-movement training induced significant altered activation to brain activity in the non-affected, contra-lesional, and extra-striate cortical areas [38]. Thus, stipulation by the Masuda et al. study that improvement in visual LPZ activity is not directly correlated with eye movement does not exclude the impact of eye-movement training on non-affected cortical areas.

Based on the results from the studies reviewed in this paper, the notion of neuroplasticity as an underlying mechanism by which MD patients are experiencing visual improvement does not seem far-fetched. World leading companies such as Tobii use eye-tracking technology infused with virtual reality headsets to help gamers, professional athletes, medical professionals, and engineers. Their eye tracking software helps identify their eye movement habits while performing their tasks and provides input on how one can improve upon it through augmented reality (AR) training [39]. The individuals utilizing eye tracking software and hardware by companies such as Ergoneers, Remobi, Gazepoint, SR Research, Tobii, and others are mostly healthy individuals. They are refining their visual skills and hand-to-eye coordination such as in the case of athletes and even medical professionals, to improve their reaction times and target tracking and focus. When combining the promising data from the studies reviewed in this paper and the how healthy individuals are refining their visual skills through advanced eye-tracking systems, the question of what is next arises. AR headsets coupled with advanced software by eye-tracking companies and guided by medical specialists could offer an inexpensive, non-invasive, and ultimately accessible approach to eye-movement training. It could be considered as a form of rehabilitation and therapy for those with central visual field impairment. Patients could utilize this technology in-clinic or even at home. Clinicians or low-vision therapists can customize the preset programs so that meet each patient’s needs. Furthermore, intelligent software could provide both the patient and the healthcare provider with instantaneous feedback (reading speeds, fixation, gaze tracking, saccades) on smart devices and can easily be stored and evaluated. Additionally, it could adapt automatically to a patient’s performance and increase or maintain the reading speed or magnification for instance. Having an accessible means of training means that patients could start experiencing and reporting QoL improvements in their daily tasks. If neuroplasticity is the platform on which eye-movement training is providing benefit, the outcome in medical, social, and economic improvement could determine to be worthwhile the investment.

Studies have utilized a standardized QoL assessment tool after eye-movement training include that of Coco-Martin et al. and Kaltenegger et al. 2019 that revealed a significant improvement in the QoL of patients with central vision loss after two different forms of vision training. Interestingly, there seemed to be a direct correlation between improvement in reading speeds and QoL. Reading speed improvement as revealed by this review could be achieved by a multitude of training manners and does not seem strictly bound to one training methodology. The Kaltenegger study revealed a significant decrease in depressive symptoms after an improvement reading speed was achieved. In general, combining different forms of reading training, optimized magnification, and eccentric viewing training seems to yield the best results. Improvements to QoL encompass changes such decrease in daily psychological distress and more ease in completion of visually demanding tasks. Additionally, it could indicate a less likelihood for dependence on social workers and more freedom. A decrease of depressive symptoms and increase of independence can bear healthcare benefits on the long term such as more activity and less need for mental health support. It would be interesting to see future studies that would incorporate long-term QoL assessment follow-ups. There might be tremendous advantages in combining accessible and relatively less expensive devices such AR headsets preloaded with eye-tracking and reading training software. It became more evident that implementation of eye-movement training into modern day vision rehabilitation could be crucial. In addition to improving QoL, these training practices open doors to further understanding the neuroplastic mechanisms in the brain.

## 5. Conclusions

Eye-movement training has shown overall promise in ARMD patients in particular and individuals with central vision loss in general. Improvements included reading speeds, QoL, VA, fixation, saccades synchronicity, and letter/face/object recognition. The use of augmented reality and eye-tracking technology to improve the visual skills of healthy individuals has further given indication that a this could be an accessible technology for clinics and patients alike. The connection between the benefits of eye-movement training and neuroplasticity certainly cannot be excluded. Future studies might delve into the possible neuromodulation taking place in a patient undergoing eye-movement training using novel AR aided technologies. In combining the latter with further QoL evaluations, this would aid in better understanding both the micro and macroscopic scale impact that such training can have on a patient’s life. Although various studies have shown promising results, there seems to be a lack of uniformity or conformity to a certain methodology. This would aid future researchers in choosing what methods to choose in aim of reproducing similar or better results. Finally, the correlation between eye-movement training suggested by previous studies and pioneers in eye-tracking technologies would be very interesting to further investigate. Future exploration of this correlation and the standardization of eye-movement training could be of great use in grasping the potential of vision rehabilitation and implementing it in the clinical setting. Eye-tracking AR headsets combined with eye-movement training protocols could one day be an inexpensive, non-invasive, and accessible means to improving the quality of life of many.

## Figures and Tables

**Table 1 brainsci-12-00036-t001:** Mean reading speeds (wpm) before and after eye movement training in Morales et al. 2019.

Group A Patients	Words Per Minute
Before BFT	56 ± 30
After BFT	63 ± 36
**Group B Patients**	**Words Per Minute**
Before BFT	56 ± 30
After BFT	63 ± 36

**Table 2 brainsci-12-00036-t002:** Outline of the different studies included.

Study	Experiment Type	Number of Participants	Disease Type	Length of Study	Results
Nilsson et al. (2003) [24]	Trained Retinal Locus evaluation and SLO utilization	20	ARMD	5–7 weeks	+ Reading speed and eccentric fixation
Kaltenegger et al. (2019) [25]	Rapid Serial Visual Representation (RSVP)	37	ARMD	12–18 weeks	+ Reading speed and QoL
Coco-Martin et al. (2017) [26]	Reading Rehabilitation Program	36	STGD, AFVD, MMD	6 weeks	Improvement in Reading speed and Qol
Seiple et al. (2005) [27]	Model 504 Pan/Tilt; Applied Science Laboratories [ASL]	16	ARMD	8 weeks	+ Reading Speed
Stigchel et al. (2013) [28]	Induced scotoma and visual challenges (EyeLink II)—No training involved	19	Stargardt and induced central scotoma	256 trials	MD patients had longer search times and partial central scotoma
Janssen et al. (2016) [29]	Induced scotoma and visual challenges	9	Induced central scotoma	2 weeks	No Significant Improvements
Léne et al. (2020) [30]	Induced scotoma and visual challenges (EyeLink II)	15	Induced central scotoma	10 days	+ Saccade reaction time
Morales et al. (2020) [31]	Biofeedback fixation training (BFT) with microperimetry	67	Mixed *	2 weeks	+ Reading speed and fixation
Walsh et al. (2014) [32]	Induced scotoma and visual challenges (EyeLink II eye tracker)	12	Induced central scotoma	3–6 weeks	+ Search reaction times and fixation
Liu et al. (2016) [33]	Induced scotoma and visual challenges (EyeLink II)	8	Induced central scotoma	6–10 h	+ Letter/face/object recognition, spatial attention, and reading speeds
Li et al. (2011) [35]	VA recovery in amblyopia after video game therapy	20	Amblyopia	20–40 days	Five times improvement in VA recovery vs. Occluson therapy alone

+ sign indicates improvement in the mentioned parameter. Mixed * indicates retinal geographic atrophy, dry ARMD, Best’s disease, myopic macular, and central serous macular degeneration (CSR) *n* = 3.

## Data Availability

Data sharing not applicable. No new data were created or analyzed in this study. Data sharing is not applicable to this article.

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
