# Peer review of "Benefits of Implementing Eye-Movement Training in the Rehabilitation of Patients with Age-Related Macular Degeneration: A Review"

_brainsci, 2021, doi:10.3390/brainsci12010036_

Round 1
Reviewer 1 Report
In this article the authors review studies that assessed eye movement function and training to improve the quality of life in age-related macular degeneration (ARMD) patients. This is an important topic and eye movement training is a promising tool to help patients suffering from ARMD. The authors focus on 7 studies published between 2005 and 2019 and discuss eye movement behaviour in the patient group, eye movement changes after training, and if possible, the relationship between eye movement function and patients’ quality of life.
I appreciate the level of detail the authors provide for each study. However, I believe it would be beneficial for the reader if the information given in the review was restructured by eye movement function or training benefit rather than to present each study separately. I have a few suggestions how the review could be restructured. These suggestions may not be the optimal way of synthesizing the information. I am therefore not set on “my way”. However, I do believe that the results need to be grouped by a common theme to make an accessible contribution to the field.
- There is a lot of general information about ARMD given at the beginning of the Results section. I would prefer to read about disease prevalence, risk factors, and visual and behavioural difficulties (e.g., what symptoms are related to quality of life) in the Introduction. Moreover, a short summary of different types of eye movements, their general function, and the goal of eye training paradigms should be included in the Introduction. It was surprising to me to see that only saccades were considered in all studies. Are there any studies looking at eye movements in response to moving stimuli? If not, what is the reason behind focussing on saccades?
- In the Discussion, the authors state that “The most significant improvements were in letter/face/object recognition, spatial attention, fixation, and reading speeds.” (p.10, l.466-467). This is not something the reader can easily derive by reading the Results section. I would suggest that the authors structure the Results by either grouping studies by eye movement training tasks or eye movement types (e.g., saccade vs. fixation). This way it is possible to compare the included studies directly. Moreover, I was wondering whether length and frequency of training had any effect? In the current form it is difficult to compare across studies because results are reported sequentially. A summary table may help the reader (and the authors) to clarify the common theme among the different studies.
- In the Discussion, I would like to see a bit more speculation about why eye movement training may work. This is again something that might be easier to spot if the studies are synthesized in a different way. For example, does eye movement training improve visual acuity, i.e., fixation stability, reading ability etc.? Or does eye movement training improve oculomotor function, i.e., saccade latencies, inhibition etc.? If these patients mostly benefit from improving visual acuity, maybe alternative training approaches, such as augmented reality could be even more promising. In other words, is eye movement training improving brain function, eye movement “strategy”, or information processing?
- Given the vast number of acronyms, please include a glossary. Please also make sure to be consistent. For example, “quality of life” is sometimes spelled out and sometimes abbreviated.
Author Response
Cover Letter
In the following letter, I shall address the valid and appreciated points addressed by the respected reviewer to refine the clarity, structure, and content of the submitted manuscript.
- In the introduction, epidemiological data regarding age-related macular degeneration, quality of life struggles for patients has been restructured and placed accordingly. Additionally, general information regarding eye movements, their different types as well as the goals of eye movement training paradigms has been introduced so that the informative content adds value to the text. Saccades, fixation, letter/face/object recognition and reading speeds are some of the parameters that have been tested by the authors of the reviewed studies. Although a variety of these parameters were under investigation, my analysis of these studies revealed that focus was heavily directed on training that involved fixation and saccades especially.
- In my revision of the discussion section, the reviewer provided helpful feedback. Studies were separated into sections focused on reading and quality of life and those focused on oculomotor training and fixation. I did, however, improve the presentation and clarity of the information regarding the other visual parameters tested and how they were assessed by the authors of the studies. Additionally, further elaboration regarding the significance of findings beyond saccades and fixation was weaved throughout the discussion extensively.
- Table 3 gained a new results column that summarizes the core findings of each study. Expansion to the scope of eye-movement and its underlying bridges to neuroplasticity is now an integral part of the discussion. Additionally, examples of world leading technology companies that incorporate augmented reality headsets retrofitted with eye-tracking hardware & software were provided. These examples shine light on how otherwise visually healthy individuals (athletes, first responders, police officers, gamers, medical professionals) are benefiting from eye-movement training and eye-tracking devices.
- A glossary is now placed after the abstract to avoid confusion and refine clarity.
Finally,
I would like to thank the respectable journal and the reviewer for the feedback provided and the time allocated to my paper. There is always room for improvement, and I am glad to have been granted time for mine.
Best regards, Anis Hilal
Reviewer 2 Report
Hilal and colleagues present a review on the benefits of implementing eye-movement training in the rehabilitation of patients with age-related macular degeneration, which is overall a timely and important topic to address in my opinion. However, it was a bit difficult for me to follow their line of argumentation. Some reasons for conducting this review become clearer only in the discussion at the end of the manuscript, but should in my view be picked up earlier already in the introduction in more detail.
Also, it does not become clear to me, how the authors made their selection of studies they describe in more detail in their review while discarding others. There is no reference to the Tables 1 to 3 (p. 2f.) in the text. What makes the six eye-movement studies chosen special or important? Why were other training studies, like for example Nilsson et al. (2003; DOI: 10.1016/s0042-6989(03)00219-0), Gustafsson & Inde (2004; DOI:10.3233/TAD-2004-16403), Kasten et al. (2010; https://doi.org/10.1177/0145482X1010400506), Nguyen et al. (2011; https://doi.org/10.1111/j.1755-3768.2010.02081.x), Coco-Martin et al. (2013; DOI: https://doi.org/10.1016/j.ophtha.2012.07.035), Kaltenegger et al. (2019; https://doi.org/10.1007/s00417-019-04328-9) or Chatard et al. (2020; DOI : 10.2174/1874609812666190913125705), among others, not considered?
Another question I would have is, why the authors base their argumentation concerning oculomotor behavior in ARMD patients almost solely on the study by Van der Stigchel et al. (2013). While this study is certainly interesting, its main results appear to be based on the data of only four patients with central vision loss (, who also don’t have age-related macular degeneration by the way, but a juvenile form, Stargardts’ disease, which should be corrected in the manuscript). Why not broaden the argument by also taking the results of other studies investigating visual search in patients with central vision loss into account? For example, Thibaut et al. (2017; https://doi.org/10.1111/cxo.12644) or Thibaut et al. (2019; https://doi.org/10.1111/cxo.12982) come to mind here, to name only two.
Other issues:
Figures 1 and 2: please specify the y-axes and also describe the x-axes in more detail;
l. 277: the supplementary material provided does not include a table, but a figure showing obviously again fixation stability of group B patients.
Table 4 contains the same values for both groups?
l. 465: The authors mention a “paradigm shift”. What exactly do they mean by this term?
l. 509: The authors write about the potential of training protocols to improve the patients’ QoL, but they miss to cite studies probing that very question, for example by measuring QoL before and after training measures. In l. 513 the authors write, “it does not seem common to find studies that have subjected their patients to extensive QoL assessments such as the NEI VFQ-25 QoL and others prior and post-vision training programs”. It does not become clear, if there are any nevertheless, and why they were not cited. A quick search by my own reveals at least a few. For example Kaltenegger et al. (2019; https://doi.org/10.1007/s00417-019-04328-9) and Coco-Martin et al. (2013; DOI: https://doi.org/10.1016/j.ophtha.2012.07.035) had obviously done so.
Minor comments:
l. 29: A dot at the end of the sentence is missing.
l. 43: The citation 53 does not exist?
l. 58 and elsewhere: Do you possibly mean systematic instead of systemic review?
Table 3: Year dates are sometimes in brackets and sometimes not, please be consistent.
l. 52-56 and l. 60-66: These paragraphs appear largely overlapping – what is the point of not merging them?
l. 68: I would suggest to change the subheading 3.1. to Age-Related Macular Degeneration and Quality of Life
l. 72: In that sentence a verb appears to be missing.
l. 74/75: “on the outskirts of 50” – maybe better: in the late 50s?
l. 76: “shy of” – maybe better: almost?
l. 100/101: sentence?
l. 104/105: I would suggest to delete the sentence “A patient was excluded…”, or to bring it later.
l. 165-167: Some adequate citations should be added in my opinion in regard to the development of the PRL and the “various studies” mentioned in l. 167.
l. 174: As long *as*…
l. 198: It should be William Seiple.
l. 208/209: Is there a “to be” or something like that missing in that sentence?
l. 310-335: The experiments conducted by Janssen et al. (2016), and the transfer effects, are a bit difficult to understand without further information.
l. 346: should “stimulate” be “simulate”?
l. 348: what is meant by “an ocean of the letter C”? That appears rather imprecise.
l. 336ff.: In the description of the study by Walsh et al. (2014), it is said that the authors of that study aimed to find out, what peripheral retinal locations would be chosen by the subjects. I guess this refers to upper, lower, left or right to the simulated scotoma? What were the results on that? I think that would be interesting information.
l. 366: There is some number missing in the year date.
l. 367f.: sentence?
l. 393ff.: What are the bullet points for here?
l. 380ff.: It is also not quite clear in my view, what exactly the task was in this study and how the results relate to it. For example, where on the screen was the target located on each trial? What does it mean, when there were corrective saccades in the horizontal direction?
l. 449: What is meant with “blue sessions”?
l. 516f.: sentence?
l. 536ff.: The authors mention studies on neuroplasticity in regard to eye movement trainings for the future, and by that seem to disregard that some studies on that subject already exist.
Author Response
Cover Letter for Reviewer 2
In the following letter, I shall address the valid and appreciated points addressed by the respected reviewer to refine the clarity, structure, and content of the submitted manuscript.
- I have gained insightful and very relevant knowledge from the articles you mentioned and added to the content of this review material from Nilsson et al. - Cocco-Martin et al. Kaltenegger et al. and others. I chose to include the selected articles among others based on the valuable content they added to the review.
- It was important to further broaden the scope by including content that revealed the impact of different reading training techniques on reading speed and consequentially quality of life.
- The latter allows the reader to flow through the text and grasp the message regarding the potential of eye-movement training and ultimately neuroplasticity.
- I removed unnecessary wording (such as paradigm shift), bullet points, tables (previously 1&2) and figures (1&2) that were otherwise repetitive and did not add informative value.
- I corrected spelling, grammatical errors, reference placement and numbering, and added a glossary.
- The Van der Stigchel et al. study review has been corrected and re-written completely so that it ties with the context of the text.
- The review of Walsh et al. study was completed with content regarding the peripheral locations chosen by subjects and the authors’ perspectives on the matter.
- Table 3 (now named Table 1) and has gained a Results column that summarizes the outcome of the reviewed studies individually.
- The discussion section no longer presumes that there is a lack of studies regarding QoL assessment coupled to eye-movement training. Rather, after having thoroughly reviewed more articles, it became clear that a clear connection has previously been documented. Thus, this became a point to elaborate on and expand further in the discussion.
- Many small changes and corrections ensued after your greatly appreciated feedback. I truly felt that you helped elevate the standard and content of this review.
Finally,
I would like to thank the respectable journal and the reviewer for the time and care allocated to this review. There is always room for improvement, and I am glad to have been granted time for ours.
Best regards, Anis Hilal
Round 2
Reviewer 1 Report
The authors addressed all my comments.
Thanks!
Author Response
I would like to thank the respectable journal and the reviewer for the time and care allocated to this review. You have helped improve its context and standard for the benefit of its authors and readers alike.
Best regards, Anis Hilal
Reviewer 2 Report
The manuscript has greatly improved in my view by the changes made.
The following minor points did I notice:
- The description of the study of Seiple et al. (2005) appears twice in the text.
- l. 1063ff. The introduction of the LPZ without further elaboration on its role in neuroplastic changes appears a bit out of context.
- Visual acuity measures appear to be reported throughout the text in different scales, while the scales are not always mentioned. Maybe it would help the readers to add this information, where appropriate.
Author Response
Cover Letter for Reviewer 2
Review Round 2
In the following letter, I shall address the valid and appreciated points addressed by the respected reviewer to refine the clarity, structure, and content of the submitted manuscript.
- The Seiple et al. study was described twice, and this was corrected.
- Thank you for your comment regarding the LPZ. I elaborated further on the matter in relation to neuroplasticity, eye-movements, and eye-movement training.
- I corrected for writing visual acuity in different scales and included scales for both logMAR and Snellen (feet).
Finally,
I would like to thank the respectable journal and the reviewer for the time and care allocated to this review. There is always room for improvement, and I am glad to have been granted time for ours.
Best regards, Anis Hilal